# How Management System Affects the Concentration of Retinol and α-Tocopherol in Plasma and Milk of Payoya Lactating Goats: Possible Use as Traceability Biomarkers

**DOI:** 10.3390/ani11082326

**Published:** 2021-08-06

**Authors:** Mercedes Roncero-Díaz, Begoña Panea, Anastasio Argüello, María J. Alcalde

**Affiliations:** 1Department of Agronomy, Universidad de Sevilla, Ctra. Utrera km. 1, 41013 Seville, Spain; mroncerodiaz@gmail.com; 2Unidad de Producción y Sanidad Animal, Centro de Investigación y Tecnología Agroalimentaria de Aragón (CITA), Avda. Montañana 930, 50059 Zaragoza, Spain; bpanea@cita-aragon.es; 3Instituto Agroalimentario de Aragón—IA2 (CITA-Universidad de Zaragoza), 50059 Zaragoza, Spain; 4Department of Animal Pathology, Animal Production, Bromatology and Feed Technology, University Campus of Arucas, School of Veterinary Sciences, Universidad de las Palmas de Gran Canaria, 35416 Aruca, Spain; tacho@ulpg.es

**Keywords:** biomarker, traceability, goat, carotenoids, retinol, α-tocopherol

## Abstract

**Simple Summary:**

The milk production systems in goats are mainly intensive and semi-extensive. In the former, the goats are housed and are fed a total mixed ration with vitamin supplementation. In the latter, the feeding of the goats is based on grazing, although with some supplementation with compound feed. Retinol and α-tocopherol that appear in these animals come from the feeding regime, since the animal cannot synthesize them. The objective of this study was to verify if the vitamins provided in different management systems of Payoya lactating goats are good markers of the production system. For this purpose, the content of carotenoids, retinol and α-tocopherol in the milk and plasma of the goats was quantified. Results showed an inverse relationship of the amount of these vitamins between milk and plasma. On the other hand, the metabolism of different vitamins depends on their origin (natural/synthetic), with natural α-tocopherol and synthetic retinol showing the highest amount in milk. Finally, it was found that these compounds can be good traceability tools which allows to guaranty to the consumer the origin of the products derived from these animals.

**Abstract:**

The retinol and α-tocopherol concentrations were quantified (μg/mL) using high-performance liquid chromatography (HPLC) in both plasma and milk of goats from three management systems. The aim was to investigate if the compounds pass from feed to animals’ fluids and to evaluate their potential use as feeding regime biomarkers. A total of 45 Payoya dams were distributed in three groups according to management system during the first month of lactation: mountain grazing (MG), cultivated meadow (CM) and total mixed ration (TMR). TMR group had higher concentrations of retinol in both plasma (25.92 ± 3.61 at 30 days postpartum) and milk (8.26 ± 0.79 at 10 days postpartum), and they were also the unique animals whose milk contained detectable concentrations of α-tocopherol (3.15 ± 0.19 at parturition). However, MG and CM goats showed higher plasma concentrations of α-tocopherol (64.26 ± 14.56 and 44.65 ± 5.75 at 30 days postpartum, respectively). These results could imply differences in the bioavailability of supplemented vitamin A and natural β-carotene and between the natural/synthetic forms of α-tocopherol. An inverse relationship between the fluids (plasma/milk) in the contents of α-tocopherol and retinol was observed as lactation progressed. Since 80% of the animals were correctly classified using a discriminant analysis based on these vitamins, these compounds could be used as traceability biomarkers of feeding system, but further studies are necessary to know the possible passage to kid meat.

## 1. Introduction

Although the European Union (EU-28) had only 1.1% of the world goat population as of 2019 [1], the EU produces 12.3% of the goat milk produced worldwide. Spain is the second country in the EU in terms of the goat census and goat milk production (22% of whole UE production). Southern Spain (Andalusia) has 37% of the Spanish goat census and a production of approximately 220,000 tons of milk per year, representing 42% of total Spanish production. Around 91% of this goat milk is transformed into cheeses [2] that have regional or local connotations of origin and quality, which is very important since consumers demand this information [3]. Therefore, advances in food traceability (techniques and technologies) facilitate accurate authentication throughout the food chain.

In Spain, goats are raised in different systems. The most specialized breeds are usually raised in confinement, feeding on cereals. However, in the most disadvantaged areas, goats are often raised in silvopastoral systems, with concentrate supplementation when pastures are of poor quality or scarce. Goats are animals specially adapted to marginal areas that other species cannot use, so their breeding is highly sustainable in environmental terms [4]. In addition, the use of these marginal areas fixes population in the territory, helping social sustainability. Therefore, although silvopastoral systems are less efficient economically [5], this less intensified traditional management is amply justified [6,7].

The Payoya breed is an autochthonous Spanish breed listed as a special protection breed, endangered [6]. This breed originally had a dual purpose of milk and meat production, but in recent years, it has become selected for milk production due to its high production and cheese quality [7]. This local breed is rustic and versatile that allows both semi-intensive and semi-extensive production systems. It is fully adapted to the environment in which it lives, including marginal areas of difficult access [8]. Most of Payoya breed farms are raised under semi-extensive system based on the grazing of natural pastures and the main difference between them is the concentrate consumption per animal/year [9] depending on the farmlands conditions.

The sustainability of these traditional rearing systems [4] and the milk and meat quality of the kids produced on them, were taken into account to find a tool to discriminate how the animals were reared could be a useful external cue of quality for the consumers concerned about the relationships between environment, animal production and food quality [10].

Carotenoids are lipophilic pigments found in plants [11]. Some carotenoids are precursors of retinol (or vitamin A) [5], with β-carotene being the main provitamin A carotenoid [12]. On the other hand, tocopherols (or vitamin E) are found naturally in leaves, green parts of plants, seeds and vegetable oils (such as soy, corn, sunflower or cotton) [13], with α-tocopherol being the most important form. These compounds cannot be synthesized either by mammals [6] or by rumen microorganisms [14]. Therefore, the presence of these compounds in animal fluids and tissues is always derived from the animal’s diet, and for this reason, they can be used as traceability tools [15,16]. Nevertheless, from the best of our knowledge, there is little literature concerning the presence of these compounds in goat fluids and tissues. Fedele et al. [17] concluded that retinol concentration on goat milk depends on the presence of provitamin A or their precursors in the animals’ feed. Pizzoferrato et al. [18] studied the content of vitamin E and their antioxidant activity to establish if goats were reared based on pasture or based on cereal. More recently, Delgado-Pertiñez et al. [19] and Gutiérrez-Peña et al. [20], studying concentrations of vitamins E and A in goats reared in different grazing systems, stated that both vitamins could be used to determine the season in which the milk was produced.

Therefore, the objective of the present study was to know about the influence of three management systems (mountain, meadow and indoor) and quantify the concentration of retinol and α-tocopherol in the plasma and milk of goats reared in these management systems, as the previous step to establish if any of these compounds should be or not search in the kids’ meat.

## 2. Materials and Methods

### 2.1. Geographical Location and Animal Management

All animal management was conducted according to the conventional management in farms and to the guidelines of Directive 2010/63/EU on the protection of animals used for experimental and other scientific purposes [21].

The study was conducted in the province of Cádiz in southern Spain on three goat farms. Two of them were in the municipality of Grazalema (36°47′56.43″ N, 5°19′57.91″ W and 36°44′45″ N, 5°24′21″ W), and the third was in the municipality of El Bosque (36°43′47″ N, 5°30′47″ W). The Sierra de Grazalema Natural Park is the area with the highest rainfall on the Iberian Peninsula [22]. It is characterized by a semiarid Mediterranean climate, with an average annual precipitation of 2093.92 mm/m^2^/year [23] with seasonal distribution from October to April, wet and cold winters, and dry, hot summers. Thus, the climate offers high- and low-production grazing periods, with high-production periods usually coinciding with spring and autumn and low-production periods coinciding with summer [19].

A total of 45 goats from the Payoya breed were selected between their third and sixth parturitions, all of them weighed between 50 and 60 kg of live weight. The amount of forage ingested by each animal was not controlled, as they presented a good body condition due to a proper compound feed ingested that was adapted to the quality and production of the pasture.

The study was developed in spring. Three groups of 15 animals each were established according to the management system. The description of the feeding regime of the different management system is detailed in Table 1.
(a)Mountain grazing (MG). Goats grazed ad libitum for eight hours per day in a pasture composed of 60% *Poaceae* (*Gramineae*) and other families. The rest of the time, they were stabled and received 800 g/day of cereal-based pellet compound feed. The farm was located at 934 m of altitude and mid-mountain weather conditions.(b)Cultivated meadow (CM). Goats grazed ad libitum for eight hours per day on a pasture composed of 95% *Poaceae* (*Avena sativa*) and other families. The rest of the time, they were stabled and received 500 g/day of compound feed composed of 28.45% MG pellet compound feed + 35% sunflower seed (*Helianthus annuus*) + 28.3% oat seed (*Avena sativa*) + 8.25% pea seed (*Pisum sativum*). The farm was located at 298 m of altitude and valley weather conditions.(c)Total mixed ration (TMR). Goats were always stabled and fed 1.5 kg/day total mixed ration based on cereal and hay ad libitum.


(a)Pasture. Three sampling times were considered: at the time of parturition (M1), at 10 days of lactation (M2) and at 30 days of lactation (M3). Samples were collected according to the procedures of Fernández et al. (1993) [24]. Briefly, samples were collected in triplicate, classified by botanical families, weighted and percentages of each family were calculated. Then, samples were stored in self-sealing transparent bags labeled and identified with the botanical family and the sampling date and thereafter placed in a freezer at −80 °C for subsequent lyophilization and analysis.(b)Pellet compound feed. Samples were obtained directly from commercial bags and placed in transparent self-closing bags labeled with the farm identifier and the sampling date. Then, the samples were stored in a freezer at −80 °C until analysis. Following commercial practices, compound feeds were supplemented with vitamins A and E using encapsulation technology.(c)Hay. Hay was provided only to TMR goats. Samples were taken directly from the hay bales on the farm, placed in self-sealing transparent bags and labeled with the sampling date. Then, they were stored in a freezer at −80 °C for subsequent lyophilization and analysis.


### 2.2. Sampling of Blood and Milk

Three sampling times were considered: at the time of parturition (M1), at 10 days of lactation (M2) and at 30 days of lactation (M3). Sampling was always carried out at the same time, at 8:00 a.m.
(a)Blood. For each sample, 10 mL was extracted from each goat in duplicate by jugular puncture and collected in VACUTAINER vacuum tubes. Li-heparin was used as an anticoagulant. The samples were labeled with the goat’s identification code, the farm and the date. The samples were cold transported to the laboratory at 4 °C in a portable icebox. The samples were centrifuged (2500 rpm, 10 min, 4 °C) in an Eppendorf 5810 R centrifuge to separate the plasma, which was stored at −20 °C in the freezer until analysis.(b)Milk. Two 30 mL samples of milk from each goat were collected aseptically in Falcon tubes and protected from light. The samples were taken in the morning before milking after eliminating the first jets. The samples were labeled with the goat’s identification code, the farm and the date. The samples were cold transported to the laboratory at 4 °C in a portable icebox. The samples were kept in a freezer at −20 °C until analysis.

### 2.3. Extraction of β-Carotene, Lutein, Retinol and α-Tocopherol


(a)Feed. Extraction of the fibrous and concentrated feed samples was performed in duplicate according to the methodology described by Pickworth et al. [25]. Samples were homogenized with the aid of a mortar, and the solvent was slowly mixed with 5 g of sample until coloring ceased to form. The solvent used for the extractions was a mixture of hexane and ethanol (1:1, *v*/*v*). The resulting liquid was filtered and then saponified overnight at room temperature in darkness and in a nitrogen atmosphere with a potassium methanol solution of 10–15%. Subsequently, the liquid was transferred to a separatory funnel and distilled water, and 1 mL of diethyl ether was added. After the phases were separated, the aqueous phase was discarded. The organic phase was washed with demineralized water until neutralization. Subsequently, the resulting extract was evaporated with a stream of nitrogen gas, and the dry residue obtained was resuspended in 100 μL of ethyl acetate for subsequent analysis by high-performance liquid chromatography (HPLC).(b)Plasma and milk. The compounds were extracted from plasma samples using the protocol described by Lyan et al. [26] and for the milk samples, the protocol described by Nozière et al. [11] was used. Both extractions were performed in duplicate for each sample and shared the initial stages of the procedures. For each extraction, 2 mL of sample, 1 mL of distilled water, 2 mL of ethanol and 2 mL of hexane were mixed in 50-mL Falcon tubes. The organic phase was separated with Pasteur pipettes after centrifugation (2500 rpm, 10 min, 4 °C). In the aqueous phase, hexane extraction was repeated two more times. On the one hand, the extracts obtained from plasma were evaporated to dryness with a stream of nitrogen gas. The dry residue was resuspended in 50 μL of ethyl acetate for subsequent analysis by HPLC. On the other hand, the extracts obtained from milk were saponified with 15% ethanolic potassium hydroxide overnight at room temperature in darkness and in a nitrogen atmosphere to hydrolyze the triacylglycerols integrated into the milk fat [27]. The next day, the samples were washed by adding distilled water to remove excess potash and subsequently evaporated to dryness using a stream of nitrogen gas. The resulting dry residue was resuspended in 50 μL of ethyl acetate for HPLC analysis.


### 2.4. Chromatographic Conditions

HPLC analysis was performed on a Varian Pro STAR 240 system equipped with a photodiode detector, a quaternary pump, a temperature control module set at 20 °C, an automatic injector (injection volume of 20 μL) and a Hypersil ODS C18 column (150 mm × 4.6 mm, 5 μm). The elusion gradient described by Mouly et al. [28] was used with some modifications [29]. The mobile phase was composed of methanol (MeOH), methyl-tert-butyl-ether (MTBE) and water added in different proportions over the measurement time according to the following gradient: 0 min: 90% MeOH + 5% MTBE + 5% water; 12 min: 95% MeOH + 5% MTBE; 25 min: 89% MeOH + 11% MTBE; 40 min: 75% MeOH + 25% MTBE; 50 min: 40% MeOH + 60% MTBE; 56 min: 15% MeOH + 85% MTBE; 62 min: 90% MeOH + 5% MTBE + 5% water. The mobile phase was pumped at 1 mL/min, and the chromatograms were monitored at 450 nm for carotenoids, 325 nm for retinol and 280 nm for α-tocopherol. Every day at the end of the analysis, the column was washed with MTBE:MeOH (50:50) for 20 min. The compounds were identified using standards (lutein (PHL89723), β-carotene (C4582), all trans-retinol (R7632) and α-tocopherol (T 3251)) from Sigma Aldrich [30] (Madrid, Spain). Standards of carotenoids and fat-soluble vitamins with a degree of purity >90% were used. Quantification was performed using standard solution calibration curves.

### 2.5. Statistical Analysis

For the statistical analysis, IBM SPSS Statistics 25.0 software for Windows (March 2017) was used. For feed, plasma and milk, a general linear model (GLM) was performed, including management systems (MG, CM and TMR) and sampling time (M1, at parturition; M2, at 10 days of lactation; and M3, at 30 days of lactation) as fixed effects. No significant interaction was found between management system and sampling time; thus, these effects will be shown separately.

In all cases, significant differences between means (±standard error) were determined by a post hoc Tukey test, with a significance level of *p* < 0.05.

Finally, a stepwise discriminant analysis with cross-validation was carried out using the retinol concentrations in plasma and milk and α-tocopherol in plasma as variables.

## 3. Results and Discussion

### 3.1. Feed (Pasture, Hay and Pellet Compound Feed): Effect of Management System

Table 2 shows the means and standard errors of the concentrations of β-carotene, α-tocopherol and lutein in the different groups. Since there were no differences among the three sampling times for a given pasture, to simplify the text, only global means for composition are presented. Significant differences were found (*p* < 0.001) among the three groups for the three compounds. Regarding fibrous feeds of the management system (pasture or hay), the β-carotene contents were 495.334 μg/g, 164.319 μg/g and 2.063 μg/g for CM, MG and TMR, respectively. The α-tocopherol content was 147.839 μg/g in MG and 37.047 μg/g in CM and was not detected in the TMR group. The lutein contents were 139.327 μg/g, 54.887 μg/g and 1.000 μg/g for CM, MG and TMR, respectively. That is, the MG and CM groups had higher β-carotene and α-tocopherol contents than the TMR group. The results obtained for the TMR group, considering that the lonely fibrous feed was hay, are similar to those reported by other authors [31], indicating that the carotenoid content tends to decrease after harvest due to handling and storage as a result of oxidation caused by exposure to sunlight. The carotenoid content can be reduced by 70–90% in hay and by 40–60% in silage [11]. Other factors affecting the carotenoid content of a plant are the botanical species, moisture content, leaf/stem ratio, time of year or time of day of harvest, temperature and ambient light [32]. On the other hand, the concentration of α-tocopherol depends on the maturational state of the plant, the temperature, the storage time and the proportion of leaves in the foraged and harvested plants [11]. In our experiment, all samples were taken at the same time of year (same weather, temperature and hours of light) and at the same time of day, so the differences in the β-carotene and α-tocopherol contents found between the MG and CM pastures can be attributed to the distinct composition of the botanical species in each of the management systems.

Concerning the different compound feeds used, the natural carotenoid and α-tocopherol contents in the three management systems were lower than expected. Some authors [11,32] indicated that the content of natural carotenoids and α-tocopherol in feed is naturally low, possibly because during the manufacturing process, the pellet compound feed is subjected to heat, which degrades these compounds. It is precisely this lability of vitamins that justifies microencapsulation in commercial concentrates because this process protects molecules from degradation, improves stability and solubility and therefore can increase bioavailability and release, ensuring the vitamin intake of goats [14,33,34]. A study conducted in cattle [35] demonstrated that microencapsulation technology protects vitamins from different factors that decrease their bioavailability, such as ruminal hydrolysis, temperature, oxidation, interaction with other compounds in the diet or components present in the matrix.

To explain the relationship between the amount of the different compounds ingested by goats and the amount of these same compounds found in plasma and milk, some different interactions should be considered. In cows, it has been shown that between 40% and 70% of the β-carotene present in the pasture disappears in the rumen [36]. On the other hand, the intestinal values of α-tocopherol can reach 40% in primarily grain-based diets because this vitamin is oxidized in the presence of polyunsaturated and mineral fatty acids [36]. In the intestinal lumen, the absorption of β-carotene, retinol and α-tocopherol depends largely on the quantity and quality of the fat in the baseline diet [35] and is linked to the formation of micelles by the action of pancreatic lipase and bile salts [14]. The bioefficacy of β-carotene in the intestinal lumen decreases due to the presence of fiber or other carotenoids [37] and the differences in activity between species of enzymes (necessary for the transformation of β-carotene into retinol) [14,37,38] such as retinol dehydrogenase and retinol reductase (β,β-carotene 15, and 15 monooxygenase 1, CCO1). In addition, it is known that in goats, the conversion of β-carotene to retinol is more efficient than in cows [39].

Regarding α-tocopherol, it must be considered that the natural isomeric form of vitamin E is RRR α-tocopherol (found in plants, vegetable oils and in unenriched feeds) [14,36], while supplementation is performed with all-rac-alpha-tocopheryl acetate. Both forms of vitamin E, natural and synthetic, are absorbed by the intestine indiscriminately [14]. However, according to some authors of studies performed in cattle [40], natural vitamin E provides 2 or 3 times more α-tocopherol than synthetic vitamin E because the αTTP protein that transfers α-tocopherol from the liver to tissues is specific to the RRR form of α-tocopherol. This reasoning concerning bioavailability could explain why in the current experiment, TMR goats had higher concentrations of stored retinol (milk, plasma) and lower concentrations of α-tocopherol than pasture-fed goats (MG and CM groups).

### 3.2. Goat Plasma and Milk: Effects of Management System and Sampling

β-Carotene, retinol and α-tocopherol in animal fluids and tissues always come from the feeding regime of the animal. Some studies [15,41,42] indicate that animals fed a pasture-based diet with a high content of β-carotene and α-tocopherol have higher concentrations of retinol and α-tocopherol in their fluids and reserves (plasma, milk, fat), although the studies mentioned did not use compound feed supplemented with vitamins, unlike the current experiment.

In goats, 57.7% of the carotenoids in the bloodstream are mainly associated with the VLDL + LDL fraction (very low density lipoproteins + low density lipoproteins) [43]. However, α-tocopherol is preferentially incorporated into VLDL and is absorbed through LDL receptors in other tissues [34,44]. After metabolization by the liver, retinol and α-tocopherol appear in the plasma [44].

#### 3.2.1. Plasma

Results are shown in Table 3. Neither lutein nor β-carotene was detected in the plasma of any of the three management systems studied. This finding partially agrees with the results obtained by Yang et al. [43], who did not detect β-carotene in goat plasma, although they detected trace amounts of lutein in goat serum.

Independent of the sampling time, the retinol content was higher in the TMR goats than in those fed mainly pasture-based management (*p* < 0.001), which may be due to differences in the amount of this vitamin provided in the three management systems, as indicated above, and the increased bioavailability of retinol (provided encapsulated) [45,46] compared to β-carotene [15] (provided in fibrous feed) under the experimental conditions. As exposed in the review of Sauvant et al. [46], vitamin A with trans double bonds in the isoprenoid side chain undergoes degradative reaction characteristics of conjugated double bonds, which result in the partial or total loss of vitamin A bioactivity. Nevertheless, when retinyl acetate was encapsulated in ethyl cellulose, it was not degraded, passing through the four stomachs of ruminants. This is because it takes a longer period of time for ruminants to digest the cellulosic material used as the envelope [46,47]. Then, when vitamin A reaches the intestine, it is absorbed. TMR group who ingest compound feed supplemented with encapsulated vitamin A (10.000 IU), had plasma retinol concentrations much higher than pasture-based management goats. Parenteral or intravenous vitamin A administration is known to be more efficient than oral supplementation since this vitamin is destroyed in the rumen and abomasum [48]. In our study, the concentrations obtained follow the same pattern as the results obtained by Donoghue et al. [49] in sheep supplemented intravenously with high doses of vitamin A (12.000 IU) having concentrations six times higher than a control group. It is very likely that microencapsulation effectively protected vitamin A and consequently increased its bioavailability. The values found in our experiment were higher than those obtained by Yang et al. [43], who described a retinol value of 0.35 μg/mL in plasma in grazing goats.

There was no effect of sampling time on retinol content for either the CM group (*p* = 0.086) or the TMR group (*p* = 0.390), whereas for the MG group, retinol content was significantly lower (*p* < 0.001) at M1 (day of parturition) than at M2 (10 days of lactation) or M3 (30 days of lactation) sampling. Although the differences were not significant, the same trend was shown in the remaining management systems (Figure 1). Some studies [44,50] show that maternal plasma concentrations of retinol decrease from the end of gestation to parturition, reaching minimum levels around birth, as it favors the passage of nutrients to the colostrum, even in diets poor in carotenoids. According to Coelho [51], birth is a physiological state of stress during which goat retinol requirements are increased and the organism redirects retinol to meet these priority requests. Once this period has passed, plasma concentrations increase throughout the lactation period, as occurred in our study.

Some authors, [9,52] stated that plasma retinol concentrations depend on the diet composition received by the animals in the 13 days prior to sampling, and the concentration increases linearly with the daily intake of carotenoids. Following this reasoning and considering that in the current experiment the pasture composition did not change over the trial period, in the M1 samples, the plasma retinol levels are a consequence of the feed received by the goats during pregnancy, while in the M2 sampling, they are due to the feed received around the parturition, and in the M3 sampling, they correspond to what the goats are eating at that moment.

In MG, α-tocopherol was not detected at either M1 or M2 (Table 3), perhaps because of the high immunological requirements of the goats from this group during the perinatal period [14,15], which are increased by adverse weather (farm located in mid-mountain) compared to CM and TMR (farms located in valley). In this way, Blagojevic et al. [53] indicated that cold temperature increases oxidative stress and causes alteration of the immune system, and antioxidants (such as tocopherol) are necessary to mitigate its effect. In addition, all-rac-alpha-tocopheryl acetate supplementation in compound feeds (TMR goats took all the vitamin in this way) demonstrates less bioavailability than natural RRR-alpha-tocopherol due to the different efficiencies of the isomeric forms of the ester [44].

No differences between management systems were found at either M1 or M2, probably due to the expected high variability of data, as already indicated in other studies [16,41,54]. Nevertheless, at M3, α-tocopherol contents were lower in the TMR group than in the other two groups (*p* < 0.001).

The concentrations of α-tocopherol increased in plasma with the progression of lactation, which may be due to a decrease in the needs for this vitamin during the perinatal period [14,35] after parturition.

In the CM animals, M3 sampling (*p* < 0.001) presented higher values than M1 or M2 sampling, whereas in the TMR goats, the concentrations increased from M1 to M3 sampling (*p* = 0.001).

#### 3.2.2. Milk

Table 4 shows the retinol and α-tocopherol values in milk as a function of management system and sampling time. The retinol concentrations found in milk in TMR goats were higher than those in CM and MG goats (forage-based diets). These results disagree with several studies conducted with cow, goat or sheep milk [15,55,56] that showed that milk from animals fed forage-based diets has a higher retinol content than milk from animals that consume concentrate-based diets. However, some studies [57,58] concluded that vitamin A supplementation in the maternal diet during pregnancy significantly increases vitamin A in the colostrum and milk during the first days of lactation, which could explain why, in the current study, the retinol concentration was higher in the TMR group at M1, since goats from TMR group was supplemented always (before and after parturition), whereas the other groups were supplemented only after parturition.

The intake of vitamin A and β-carotene in the goats’ diets during pregnancy produced an increase in maternal liver reserves. Subsequently, these reserves were transferred from the mothers to their kids through colostrum to satisfy the kids’ retinol requirements at birth and promote the increase of their liver reserves [14,59]. Therefore, at the beginning of lactation (M1), higher retinol concentrations were obtained for all the groups studied. Around the time of parturition, the receptors involved in the uptake of retinol (RBP transport proteins) significantly increased in the mammary gland, allowing greater absorption of this vitamin and passing into the colostrum [44]. As lactation progresses, this vitamin decreased in milk. Studies conducted with cows and sheep [60,61] corroborate the presence of a higher concentration of retinol in colostrum than in milk.

In general, in the three groups of goats, retinol concentrations in milk showed a gradual decrease throughout the lactation period. There was a significant effect of sampling time in the CM (*p* = 0.019) and TMR (*p* < 0.001) goats, but there was no effect of sampling time in MG goats. In TMR goats, the decrease was more pronounced than with the CM group, in which the retinol content decreased considerably between M1 and M2 but remained constant later. The concentration of β-carotene in grass in the MG group was lower than that in the CM group (Table 2), which could explain the difference between the two groups, as happened in plasma.

There were only significant differences between management systems in M2 (at 10 days postpartum) (*p* < 0.001), while for M1 (at delivery) or M3 (at thirty days of lactation), no significant differences were found, which may also be due to the high standard error of data, which agrees with results reported by several authors regarding the absorption, circulation and storage of these vitamins in the fluids and tissues of animals [15,41,54].

The retinol values obtained in our study at the M3 sampling agree with those reported by Gentili et al. [61] for pasture-fed goats (4.3 μg/mL). In addition, Fedele et al. [17] found values between 4.82 and 6.33 μg/mL for different management systems in dairy goats, with the retinol content being higher in goats fed exclusively pasture. On the other hand, the values found in our experiment were higher than those obtained by *Kondyli* et al. [55] in pasture-fed goats during the winter months (0.013 μg/mL) and those obtained by Gutierrez-Peña et al. [20] according to grazing level and season.

α-Tocopherol was only detected in the milk of the TMR goats (Table 4), possibly because these goats were always fed compound feed supplemented with vitamin E, resulting in higher ingestions of vitamin E than the other groups. In addition, it is known that parturition creates stress, so the vitamin E needs of dams increase due to immune system activity [14,35]. TMR group goats were permanently housed, and therefore, their immune system required less vitamin E, so that the excess appeared in milk; while MG and CM groups in the last third of gestation, having a lower ingestion capacity, consumed low amounts of pasture and therefore low amounts of vitamins. Some studies conducted in sheep, goats and cows [56,62,63,64] describe higher concentrations of α-tocopherol in the milk of forage-fed animals. However, some studies in sheep fed diets supplemented with vitamin E [65,66] show that the concentration of α-tocopherol increases as a function of the supplemented dose, as occurred in our study.

Significant differences in α-tocopherol concentrations were observed among the three sampling times (*p* < 0.001). As lactation progressed, the concentration of α-tocopherol in milk decreased, in agreement with other studies in sheep’s milk [65] demonstrating that the concentration of α-tocopherol was higher in colostrum than in milk.

The α-tocopherol values obtained in our study in the TMR group were higher than those obtained by Delgado-Pertíñez et al. [19] (0.41–0.51 µg/g) and Gutiérrez-Peña et al. [20] in the Payoya breed reared in pasture but lower than those presented by Gentili et al. [61] (6.645 μg/mL) for pasture-fed goats.

#### 3.2.3. Relationship between Fluids (Plasma/Milk)

When comparing the results found for plasma and milk (Table 3 and Table 4), it can be observed that as the days of lactation progress, the concentrations of retinol in plasma increase, while the concentrations of retinol in milk decrease (Figure 1), which is in agreement with results described in cows [14]. Even though there were three very different management systems, the pattern is the same in the three management systems, therefore the average of the different systems is presented. It seems that there is a mechanism in the goat by which, and depending on the needs of the animal, retinol passes to one for another fluid. The vitamin A requirements of dams increased in the lactation period in comparison to pregnancy [61], so the retinol content increased in plasma and decreased in milk.

Similarly, there is a tendency toward an increase in plasma concentrations of α-tocopherol as lactation progresses (only in the TMR group, as it is the only group where this compound has been detected). In cows, it has been described that the requirements of vitamin E depends on the lipid fraction of the ration and its relationship with the immune system and inflammatory activity, with an increase in the necessities when animals are exposed to infectious agents [14]. Similarly, it could be hypothesized that the requirements of the goats in the perinatal period conditioned the plasma concentrations of α-tocopherol, with the concentrations lowest at the time of parturition. To the authors’ knowledge, this study represents the first time that this relationship has been revealed in goats.

### 3.3. Discriminant Analysis

The results of the discriminant analysis are shown in Table 5. Retinol concentrations in plasma and milk and α-tocopherol in plasma were introduced as variables. The analysis to correctly classify all goats used the following variables in the following order: retinol concentration in milk at 10 days of lactation, concentration in plasma during parturition and α-tocopherol concentration in plasma at 30 days of lactation. A total of 83.3% of the MG goats were correctly classified, while 16.7% were incorrectly classified to the CM group. Only 75% of CM goats could be correctly classified in their group, while the remaining 25% were classified in the MG group. Therefore, there is an exchange of animals between forage-based management systems. On the other hand, 81.8% of the TMR goats were correctly classified in their group, while the remaining 18.2% were classified in the MG group.

Therefore, based on these data, we can state that retinol concentrations in plasma and milk and α-tocopherol in plasma could differentiate 80% of animals according to the goat management systems.

## 4. Conclusions

According to our experimental conditions, during the first month of lactation, there was an inverse relationship between fluids (plasma/milk) and the content of α-tocopherol and retinol; that is, when the concentrations in one fluid decreased, the concentrations in the other fluid increased. Therefore, the amount and presence of these compounds in the two fluids is influenced by the needs of the goats in parturition and peripartum. This information makes it possible to adapt the management of these animals in these phases.

For the practical enforcement in the feeding management of goats it must be taken into account that encapsulated retinol added to the feed is more bioavailable by the animals than the β-carotene intake from grass, whereas the opposite happens for α-tocopherol. Therefore, it is relevant to highlight the importance of including encapsulated retinol in the diet of goats to ensure the requirements of this vitamin, while to ensure a correct supply of α-tocopherol, it is important that goats consume grass rich in this vitamin.

Further studies are needed to elucidate the metabolism of these compounds in goats, to clarify the bioefficacy of β-carotene and its bioavailability and to establish the bioavailability relationship between sources (natural/synthetic) of α-tocopherol.

As can be seen in the different results presented and particularly in the discriminant analysis, differences found between management systems and sampling times in both the retinol and α-tocopherol contents indicate that these compounds could be used as management traceability tools. Based on this information, it is interesting to continue researching on the passage of these compounds to the meat of kids; thus, further studies are necessary.

Finally, it is necessary to investigate if current conclusions can be extrapolated to other production systems and breeds.

## Figures and Tables

**Figure 1 animals-11-02326-f001:**
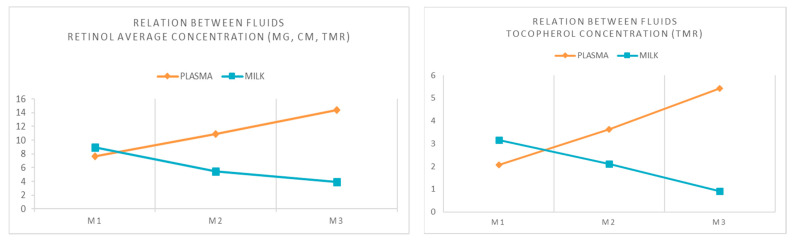
Inverse relationship between fluids (plasma and milk): Retinol average concentration (μg/mL) of the three management systems (MG, CM, TMR) and α-tocopherol concentration (μg/mL) in the TMR group at three sampling times. Management systems: MG (mountain grazing + feed supplement), CM (meadow pasture + feed supplement) and TMR (total mixed ration). Sampling times: M1 (the day of parturition), M2 (at 10 days of lactation) and M3 (at 30 days of lactation).

**Table 1 animals-11-02326-t001:** Description of the feeding regime in the three management systems of goats: feeds composition and goat consumption.

	Mountain Grazing (MG)	Cultivated Meadow (CM)	Total Mixed Ration (TMR)
Pasture (% of each botanical family)	ad libitum	ad libitum	none
	*60% Poaceae* *11% Fagaceae* *8% Geraniaceae* *6% Malvaceae* *6% Fabaceae* *5% Asteraceae* *4% Others*	*95% Poaceae* *3% Asteraceae* *1% Rubiaceae* *1% Malvaceae*	
Hay	none	none	ad libitum
			33% *Avena sativa* 33% *Vicia sativa* 33% *Hordeum vulgare var hexastichon*
Compound feed	800 g/day	500 g/day	1.5 kg/day
	*Graminae*: maize, oats, barley, wheat bran. *Legumes*: broad beans, field beans, sweet lupins, Oilseeds: genetically modified soybean, sunflower seeds.	28.45% MG concentrate, 35% sunflower seed, 28.3% oat seed, 8.25% pea seed.	*Graminae*: genetically modified maize, barley, triticale, wheat bran, oatmeal, beet pulp, genetically modified soy peel. *Legumes*: broad beans, field beans, sweet lupins, fava bean, kidney beans Oilseeds: sunflower seed, genetically modified soybean.

**Table 2 animals-11-02326-t002:** The means and standard errors of the β-carotene, α-tocopherol and lutein concentrations (µg/g DM ± SE) in three different goat management systems (GLM analysis): mountain grazing (MG), cultivated meadow (CM), and total mixed ration (TMR).

		β-Carotene	α-Tocopherol	Lutein
Mountain grazing (MG)	Pasture	164.319 ^b^ ± 2.011	147.839 ^b^ ± 2.915	54.887 ^b^ ± 1.428
	Compound feed *	2.150 ^y^ ± 0.006	17.893 ^y^ ± 0.020	0.195 ^x^ ± 0.001
Cultivated meadow (CM)	Pasture	495.334 ^c^ ± 0.046	37.047 ^a^ ± 0.001	139.327 ^c^ ± 0.001
	Compound feed *	1.120 ^x^ ± 0.06	19.547 ^z^ ± 0.233	n.d.
Total mixed ration (TMR)	Hay	2.063 ^a^ ± 0.041	n.d.	1.000 ^a^ ± 0.003
	Compound feed *	37.640 ^z^ ± 0.017	0.250 ^x^ ± 0.006	14.022 ^y^ ± 0.001
Fibrous feeds (pasture and hay) *p*-value		˂0.001	˂0.001	˂0.001
Compound feed *p*-value		˂0.001	˂0.001	˂0.001
(*) Vitamin supplementation added to the compound feed				
Vitamin E (all-rac-alpha-tocopherol acetate)		30 UI/kg	30 UI/kg	30 UI/kg
Vitamin A (retinyl acetate)		10.000 UI/kg	10.000 UI/kg	10.000 UI/kg

Significant differences (*p* < 0.05) between fibrous feeds within a column are indicated by superscripts (a, b, c). n.d.: not detected. Significant differences (*p* < 0.05) between compound feeds within a column are indicated by superscripts (x, y, z). n.d.: not detected.

**Table 3 animals-11-02326-t003:** Mean values and standard error of retinol and α-tocopherol concentrations (μg/mL) in goat plasma in three different goat management systems and at three sampling times (GLM analysis).

	Retinol				α-TocopheroL			
	M1	M2	M3	*p*-Value	M1	M2	M3	*p*-Value
MG	1.89 ± 0.52 ^ax^	9.00 ± 1.63 ^ay^	11.80 ± 1.58 ^ay^	˂0.001	n.d.	n.d.	64.26 ± 14.56 ^b^	
CM	3.68 ± 1.52 ^a^	6.83 ± 2.26 ^a^	12.06 ± 0.60 ^a^	0.086	6.36 ± 2.81 ^x^	9.10 ± 2.95 ^x^	44.65 ± 5.75 ^by^	˂0.001
TMR	18.95 ± 3.36 ^b^	20.17 ± 4.51 ^b^	25.92 ± 3.61 ^b^	0.390	2.06 ± 0.38 ^x^	3.62 ± 0.85 ^xy^	5.43 ± 0.90 ^ay^	0.001
*p*-value	˂0.001	0.007	˂0.001		0.158	0.112	˂0.001	

Management systems: MG (mountain grazing + feed supplement), CM (meadow pasture + feed supplement) and TMR (total mixed ration). Sampling times: M1 (day of parturition), M2 (at 10 days of lactation) and M3 (at 30 days of lactation). Means in the same column with different letters (a, b) differed significantly (*p* < 0.05) between management systems. Means in the same row with different letters (x, y) differed significantly (*p* < 0.05) between sampling times. n.d.: not detected.

**Table 4 animals-11-02326-t004:** Mean values and standard error of the concentrations of retinol and α-tocopherol (μg/mL) in MILK in three different goat management systems and at three sampling times (GLM analysis).

	Retinol				α-Tocopherol			
	M1	M2	M3	*p*-value	M1	M2	M3	*p*-value
MG	6.62 ± 1.46	5.59 ± 0.34 ^b^	3.84 ± 0.51	0.092	n.d.	n.d.	n.d.	
CM	9.51 ± 2.84 ^y^	2.46 ± 0.54 ^ax^	3.86 ± 0.62 ^xy^	0.019	n.d.	n.d.	n.d.	
TMR	10.69 ± 1.23 ^y^	8.26 ± 0.79 ^cxy^	4.04 ± 0.65 ^x^	˂0.001	3.15 ± 0.19 ^z^	2.10 ± 0.40 ^y^	0.90 ± 0.28 ^x^	˂0.001
*p*-value	0.389	˂0.001	0.965					

Management systems: MG (mountain grazing + feed supplement), CM (meadow pasture + feed supplement) and TMR (total mixed ration). Sampling times: M1 (the day of parturition), M2 (at 10 days of lactation) and M3 (at 30 days of lactation). Means in the same column with different letters (a, b and c) differed significantly (*p* < 0.05) between management systems. Means in the same row with different letters (x, y and z) differed significantly (*p* < 0.05) between sampling times. n.d.: not detected.

**Table 5 animals-11-02326-t005:** Discriminant analysis: Percentage of goats correctly classified (assigned (columns) against real data (rows)) based on retinol and α-tocopherol values.

Assigned/Real Data	MG	CM	TMR
MG	83.3	16.7	0
CM	25	75	0
TMR	18.2	0	81.8

Management systems: MG (mountain grazing + feed supplement), CM (meadow pasture + feed supplement) and TMR (total mixed ration).

## Data Availability

Not applicable.

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
