# Peer review of "How Management System Affects the Concentration of Retinol and α-Tocopherol in Plasma and Milk of Payoya Lactating Goats: Possible Use as Traceability Biomarkers"

_animals, 2021, doi:10.3390/ani11082326_

Round 1

Reviewer 1 Report

Dear authors, below you can find some comments to your manuscript:

Introduction

Lines 46-52: Check this paragraph. You use “approximately” three times.

Lines 71-75: It is unnecessary to state that current research is part or a project which objective is meat.

Results

I recommend to improve tables.

Regards

Author Response

Dear authors, below you can find some comments to your manuscript:

Introduction

Lines 46-52: Check this paragraph. You use “approximately” three times.

Thank you very much for detail. It has been corrected.

Lines 71-75: It is unnecessary to state that current research is part or a project which objective is meat.

Following the reviewer's suggestions, the paragraph has been removed.

Results

I recommend to improve tables.

Thanks a lot. The interpretation of the tables has been improved, following the reviewer's suggestions. 

Regards

Regards

Reviewer 2 Report

Review of Manuscript Animals-1307844

The paper aimed at evaluating the use the content of vitamins A and E in blood and milk of goats as a marker for different production system. I have for the authors the following comments:

Major comments

I regret that my remarks about the abstract were not considered. The abstract is still not really informative and too qualitative. I would recommend again some numbers (e.g. concentration of vit in milk and plasma) to give an impression of the magnitude of the results. Better explanation of experimental design and treatments is required in this chapter. How many days were goats in experiment? Detail about feeds must be given

The introduction is still not convincing. I would expect more information about the potential of using vit for traceability and the relationship between the studied vitamins and the different systems. Both were not really addressed together

M&M, information is still missing. How long lasted the experiment? My major concern about this paper is that authors did not measure or estimate the forage intake and total feed intake. Feed intake is the determinant factor for the intake of vitamins, and therefore, affects the appearance in blood and milk. So far daily intake was not measured/reported, it is not possible to make reliable interpretations and conclusions about the relationship between the feeding systems and vit in blood and milk. Therefore, it cannot be really concluded whether these vitamins have their potential to be used as markers. I still consider that a study like this without feed intake is not admissible

Author Response

Review of Manuscript Animals-1307844

The paper aimed at evaluating the use the content of vitamins A and E in blood and milk of goats as a marker for different production system. I have for the authors the following comments:

Major comments

I regret that my remarks about the abstract were not considered. The abstract is still not really informative and too qualitative. I would recommend again some numbers (e.g. concentration of vit in milk and plasma) to give an impression of the magnitude of the results.

Following the reviewer's recommendations, we have added some data. We hope not to have problems when exceeding 200 words.

Better explanation of experimental design and treatments is required in this chapter. How many days were goats in experiment? Detail about feeds must be given.

We agree with the reviewer, that it would be advisable to put more details of the experiment in the abstract, but the rules clearly indicate not to exceed 200 words. We have concentrated the information as much as possible to comply with the rules. Although, if the editor allows us, we will expand the information that the reviewer recommends.

However, in the abstract, lines 32-33 indicate that the experiment covers the first 30 days of lactation.

The introduction is still not convincing. I would expect more information about the potential of using vit for traceability and the relationship between the studied vitamins and the different systems. Both were not really addressed together

We would have preferred, as the referee points out, to have found a greater number of papers that address the study of these vitamins in goats, in order to better compare our results. However, from the best of our knowledge, there is little literature concerning the presence of these compounds in goat fluids and tissues and even less, papers related to production systems.

That said, and with all due respect, we believe that the problem is correctly exposed in the introduction.

M&M, information is still missing. How long lasted the experiment?

Both in the summary, as in material and methods (line 143-144, in section 2.2.- Sampling of blood and milk) and in tables 3 and 4, it is indicated that the samples were taken at parturition, and at 10 and 30 days of lactation.

My major concern about this paper is that authors did not measure or estimate the forage intake and total feed intake. Feed intake is the determinant factor for the intake of vitamins, and therefore, affects the appearance in blood and milk. So far daily intake was not measured/reported, it is not possible to make reliable interpretations and conclusions about the relationship between the feeding systems and vit in blood and milk. Therefore, it cannot be really concluded whether these vitamins have their potential to be used as markers. I still consider that a study like this without feed intake is not admissible.

We agree that the knowledge of forage intake would provide us more information but, with the due respect, the lack of this information does not invalidate the study.

We will try to explain ourselves. Firstly, we assume that consumption was ad libitum in all three systems, since no restrictions were placed. So, in the real conditions in which we have worked, which are those used in the production areas, the amount of vitamins found in plasma and goat's milk comes from the sum of vitamins from the ingestion of both, forage and concentrate.

So, the amount of added concentrate is adjusted to the average quality of the ingested forage, in order to meets the goats’ requirements. To control this point, the body condition was monitored, as already indicated. And to clarify this issue that the referee indicated, we have included the following paragraph in material and methods, as we did in Alvarez et al (2015):

“The amount of forage ingested by each animal was not controlled, as they presented a good body condition due to a proper compound feed ingested that was adapted to the quality and production of the pasture”.

Álvarez, R., Meléndez-Martínez, A.J., Vicario, I.M. and Alcalde, M.J. Carotenoids and fat-soluble vitamins in horse tissues: a comparison with cattle. Animal, 9(7): 1230-1238.

On the other hand, accordingly with the INRA guidelines for goats (Agabriel, 2007), the amount of dry matter ingested in the forage can be calculated as follows:

Kg DMGrass= 0.31 + 0.0015 Live Weight + 0.26 liters produced  – 0.065 kg DMCompound feed

Taking into account that a Payoya goat produces around 3 kg of milk per day during the first 5 weeks of lactation (Delgado-Pertiñez et al., 2009), we can reduce the amount of forage ingested per animal and day: for the mountain production system it would be 1.1205 kg DM, for the cultivated meadow system it would be 1,14 kg DM and for indoor system it would be 1.1725 Kg DM.

In this way, although individual intakes were not measured, we can make an approximation to the amount of vitamin ingested.

Delgado-Pertíñez, M., Guzmán-Guerrero, J.L., Caravaca, F.P., Castel, J.M., Ruiz, F.A., González-Redondo, P. and Alcalde, M.J. 2009. Effect of artificial vs. natural rearing on milk yield, kid growth and cost in Payoya autochthonous dairy goats. Small Ruminant Research, 84 ( 1-3): 108-115

Agrabriel, J. 2007. Alimentation des bovins, ovins et caprins. Besoins des animaux-Valeur des aliments. Tables INRA 2007. Versailles Editions Quae. 330 p. ISBN 978-2-7592-0020-7.

Reviewer 3 Report

The manuscript "How management system affects the concentration of retinol and α-tocopherol in plasma and milk of Payoya lactating goats: Possible use as traceability biomarkers." enters the topics of the journal.

The paper presents for the first time in goats a study that aims to know the traceability of carotenoids and vitamins A and D, from the handling of feed that goats have to plasma and milk (in this work) and even plasma and fat in the goats (in a later work), to enhance goat milk and meat. It is carried out in Payoya herds in Grazalema, according to normal management guidelines. What sometimes makes a clear explanation of the results difficult, but on the other hand it presents what there is. And it allows taking into account some guidelines that can improve the feeding of these animals and the quality of the products produced.

It was carried out on three farms, one with sawgrass and a concentrate supplement according to needs, another in a lower area with cultivated pasture and a concentrate supplement, and another stables, with concentrate and hay.

The work shows the evolution of the vitamins in the deposits (plasma and milk) and points to clear differences in the metabolic behavior of the vitamins depending on whether they have been provided naturally or synthetically (provided encapsulated in the concentrate).

In my opinion it is a very good work of advance to establish other more concrete investigations, since with this approach and objectives there is nothing in goats. In other species there is something, which in any case are not transferable because goats, as a species, have their peculiarities. It is also a complicated job because these compounds are difficult to handle, they are very light and temperature labile and there are also very high individual variations in the deposits of each goat.

Recommendations, in my opinion

Delete line 128 "Feed sampling"

the power supply for each system is included in section 2.1.

The heading 2.3., Would now be 2.2.

2.2. Sampling of blood and milk.

I suggest:

* Mark the practical importance of the work

* Reinforce and clearly explain interactions (Figure 1) according to production system

* a proposal for changes in feeding systems

* How the results could be extended to other production systems and breeds

Author Response

The manuscript "How management system affects the concentration of retinol and α-tocopherol in plasma and milk of Payoya lactating goats: Possible use as traceability biomarkers." enters the topics of the journal.

The paper presents for the first time in goats a study that aims to know the traceability of carotenoids and vitamins A and D, from the handling of feed that goats have to plasma and milk (in this work) and even plasma and fat in the goats (in a later work), to enhance goat milk and meat. It is carried out in Payoya herds in Grazalema, according to normal management guidelines. What sometimes makes a clear explanation of the results difficult, but on the other hand it presents what there is. And it allows taking into account some guidelines that can improve the feeding of these animals and the quality of the products produced.

It was carried out on three farms, one with sawgrass and a concentrate supplement according to needs, another in a lower area with cultivated pasture and a concentrate supplement, and another stables, with concentrate and hay.

The work shows the evolution of the vitamins in the deposits (plasma and milk) and points to clear differences in the metabolic behavior of the vitamins depending on whether they have been provided naturally or synthetically (provided encapsulated in the concentrate).

 In my opinion it is a very good work of advance to establish other more concrete investigations, since with this approach and objectives there is nothing in goats. In other species there is something, which in any case are not transferable because goats, as a species, have their peculiarities. It is also a complicated job because these compounds are difficult to handle, they are very light and temperature labile and there are also very high individual variations in the deposits of each goat.

We appreciate the reviewer's assessment.

Recommendations, in my opinion

Delete line 128 "Feed sampling" the power supply for each system is included in section 2.1.

 The heading 2.3., Would now be 2.2. 2.2. Sampling of blood and milk.

Following the reviewer's suggestions, the subsection has been eliminated and the numbering of the next sections has been corrected.

I suggest:

* Mark the practical importance of the work

We thank the reviewer for judging the importance of the presented study. Without a doubt, this paper indicates guidelines to follow in the practical feeding of goats. A new paragraph has been added to the conclusions.

* Reinforce and clearly explain interactions (Figure 1) according to production system

Even there are three very different management systems, the pattern is the same: concentrations increased in plasma whereas decreased in milk. Therefore, tables 3 and 4 shown the global mean of the data. Following your suggestions, this information has been clarified in the results and discussion section.

* a proposal for changes in feeding systems

Following the reviewer's suggestions, a recommendation on feeding management in goats has been included in the conclusions section.

* How the results could be extended to other production systems and breeds

We believe that the conclusions would be transferable to different goat production systems, because the study has been carried out on the most different possible systems but further studies are necessaries. We include a sentence in conclusions.

Round 2

Reviewer 2 Report

Thanks to the authors for answering my questions and considering my suggestions. The authors have now improved considerably the quality of the manuscript. I have no comments to the manuscript.

This manuscript is a resubmission of an earlier submission. The following is a list of the peer review reports and author responses from that submission.

Round 1

Reviewer 1 Report

Dear authors, I reviewed your manuscript. Comments and suggestions can be found below:

Title

I suggest to mention the breed of the goats, due to the manuscript was summited into a special issue “Competitiveness of Spanish Local Breeds”.

Introduction

Please, establish the current state of research regarding vitamins A and E in lactating goats. Deeper information is lacking.

Line 58-60: Objetive is not related to title.

Materials and methods

Please, describe with sufficient detail the experimental design.

Line 75: Describe some characteristics of the goats: age, weight, parturitions, etc (means and standard deviation).

Lines 75-77: The second and third sentence should be as part of introduction.

Line 96: Table 1. Headling of table 1 must to be improved. Why did you include the percentage of every ingredient just for CM? Why did you analyze crude fiber? Crude fiber is not useful for ruminants. Did you analyze NDF and ADF? Why analytical composition is just for concentrate?

Line 102: Table 2 is not related with text.

 Line 116: Table 2. Insert table 2 in Results section. Can you explain why did you measure lutein concentrations? Lutein is a non-provitamin A carotenoid. According with methology, retinol was extracted from feeds too, but is missing in Table 2.

Results

Lines 195-196: This sentence is not correct. Fibrous portion of diets refers to indigestible fraction of feeds, not to an ingredient.

Lines 239-248: This information can be used as a part or introduction or discussion.

Line 378: Can you show p-value? Why don’t you analyzed the relationship between fluids from goats fed of every diet?

Reviewer 2 Report

Review of Manuscript Animals-1252725

The paper aimed at evaluating the use the content of vitamins A and E in blood and milk as a marker for production system. I have for the authors the following comments:

Major comments

The title does not fit with the aim of the study. The fact that vit A and E was evaluated as potential marker has to be included in the tittle

The abstract is not really informative and too qualitative. I would recommend some numbers to give an impression of the magnitude of the results. Better explanation of experimental design and treatments is required in this chapter

The introduction is not convincing, short and too general. The main problem was not really put in context. I would expect more information about the potential of using vit for traceability and include other previous studies with similar objectives. How can be exactly the use of these vit for traceability applied? If vit E is found in grass and some concentrate ingredients, what do authors want or can really trace? Finish the introduction with your hypothesis or expected outcomes

M&M, some information is missing and in general is not described in detail. Some results are not presented properly (e.g. Table 1 and 2). Method used for chemical composition of feed are not described. One of my major concerns is that authors did not measure or estimate the forage intake and total feed intake. Although this is difficult and use of markers is required (or sampling fields before and after grazing), the total feed intake is necessary to calculate the intake of nutrients (or vitamins) and to make reliable interpretations and conclusions about the relationship between vit intake and vit in blood and milk. Conclusions cannot be really done so far total intake of vitamins is not exactly known. I consider that a study like this without feed intake is not admissible

Results are not properly presented like in Table 2. In the discussion, although the main objective was to use vit A and e as marker for traceability, this topic is almost nowhere considered in the discussion chapter. The discussion was supported by enough literature but is in general out of focus and discusses irrelevant and general issues and not too much related to the topic of the paper.

The conclusion are more a resume of results. I would recommend to change this to real “conclusions” and recommendations. What is the main message for scientists based on your results? What would you recommend to study in the future?

Minor comments

L49: I doubt whether you can trace origin. Check! Probably you will be able to differentiate products form grazing or stabulated systems but not differentiate geographical origin

L57: but how can you trace animal’s diets if almost all components of the diets contain vit E? What can you differentiate?

L59: I doubt whether you proved how vit passed from diet to plasma and milk. Was this really your aim? Check and rewrite

L72: check the unit for precipitation! “mm/year”

L 73: And the information for El Bosque?

L75: 15 per farm? Clarify

L79: How long lasted the experiment?

L82: How was the forage composition determined? Hopefully is explained in detail later

L84: What about composition of concentrate? For CM system was provided. Why not here?

L96: In table 1: give the composition of concentrate in % for all treatments. Be consistent in the presentation of information/data. Botanical composition based on DM basis? Explain. The vit content of concentrates are analysed or declared values from producer?

L106: Was DM content of each botanical family determined?

L116: content of pasture and concentrate cannot be summed because you do not know the intake and proportion of pasture and concentrate in the total diet. Therefore, all interpretation about the p-values are wrong. Except for TMR

L117: Was the lutein measurement also the purpose of this experiment? Otherwise delete values because they are no relevant

L119: If results are presented without considering vit supplementation then is wrong. You need to show total content of vit in the concentrates

L213-215: the availability of vitamins after storage in premixes is indeed reduced but the reduction is low in absolute value and only after very long storage (between 6 to 12 months). Therefore, for me, microencapsulation or use of additives are not really justified under practical conditions

L213-217: This in general deviates from the purpose and topic of this experiment. Make it shortly or delete

L252: You use different names for the vit. Use only one term and be consistent along the manuscript

L275: sixfold compared to what?

L294: Or maybe because lower feed intake around calving?

L334-335: The liver reserves of the kids was not measured here. Delete this sentence

L410-414: These are not conclusions and instead are a resume of results. Provide clear conclusions